# From the Perspective of People with Dementia: Using Creative Qualitative Measures to Assess the Values and Opinions on Freedom and Safety among People Living with Dementia

**DOI:** 10.3390/healthcare12141412

**Published:** 2024-07-15

**Authors:** Steven van Andel, Anouk Holkenborg

**Affiliations:** IJsselheem, Engelenbergplantsoen 3, 8266 AB Kampen, The Netherlands

**Keywords:** dementia, freedom, safety, values, nursing homes

## Abstract

With the growing numbers of people living in old age, a system that sustains autonomy, dignity and freedom of movement for people living with dementia (PwD) needs to be installed. However, due to the cognitive constraints in the cohort of PwD, traditional qualitative methods of inquiry, such as interviews, are often not a good match. This study aimed to use creative qualitative assessment tools to assess the values and opinions of PwD in nursing homes on freedom and safety. Twenty-two nursing home residents with memory problems participated in this study. Important themes related to freedom and safety were identified using a diverse set of methodologies. Overall, residents had a similar view on ‘safety’, relating this to having a homely environment where people look out for you, with a good balance between busy and quiet areas and being able to retain cognitive and physical function. Values around freedom were more diverse. Residents who were still capable of logical speech generally voiced a wish for independence, making one’s own decisions and wanting to go outside. For residents who no longer spoke, freedom seemed to be experienced more through the connection to other people. These results show that PwD still have distinct personal values concerning freedom and safety, which should be given due consideration in decision-making regarding nursing home policy, thereby potentially improving the quality of life of PwD.

## 1. Introduction

In the last few decades, more awareness has developed for the well-being of people living with dementia (PwD) in terms of dignity, autonomy and preservation of personal freedoms [1,2,3,4,5,6], leading to improved living conditions for PwD. With the expected rise in PwD to 132 million in 2050 [7], the aged-care system is challenged to keep up this new standard of care for more PwD with fewer working-age people to care for them. In keeping healthcare manageable without sacrificing dignity, autonomy and personal freedom, it is important to consider the values and opinions of the cohort in question.

One such topic where the input of PwD is important is the personal balance between freedom and safety. Increasing freedom of movement can be expected to have a generally positive effect on people’s lives. According to a recent systematic review study, a semi-open-door policy can lead to more physical activation, less use of psychotropic drugs, a more social living environment and general improvement in the quality of life for people with dementia [8]. On the other hand, these benefits might come with certain costs. One common argument against increasing freedom of movement is that it might increase fall rates, as moving around and greater exposure to different environments might confront PwD with more complex and risky situations. Although few studies are known to the authors, the evidence actually seems to contradict these expectations. Allowing PwD more freedom of movement in a wander garden lowered residents’ fall rates [9]. This might be explained as more freedom of movement lowers anxiety [8], which in turn is a risk factor for falls in PwD [10]. Other negative side effects of increasing movement freedom within nursing homes are the risk of getting lost and increased personal freedom, which might lead to frustration among fellow residents of nursing homes who are not allowed full mobility [11]. Full mobility outside the nursing home comes with further risks associated with interacting with traffic [12,13] and getting into other accidents in the surrounding neighbourhood, with potentially fatal consequences [14]. In short, a trade-off exists between freedom and safety, where more freedom of movement has clear positive effects for PwD, with some indications of negative side effects as more mobility is allowed. 

As PwD often do not make decisions on their personal freedoms alone, or in many cases, these decisions are entirely made for them, it is of interest to consider how different stakeholders around PwD perceive this trade-off between freedom and safety. Popham and Orrell [15] investigated expectations of PwD and the people around them of their living arrangements. They used interviews and focus groups with PwD, family members and nursing home staff. They found four themes of importance: activities and social interactions, freedom and safety, dignity and privacy, and design and environment. Regarding freedom and safety, Popham and Orrell identified a mismatch between resident and staff/manager perspectives. Residents expressed frustration at their lack of freedom, indicating a preference to be allowed more freedom in the decision of where to sit or when to go outside. Staff and managers indicated safety to be their main concern. This apparent mismatch emphasises the importance of also gaining the resident’s opinion on a personal balance between freedom and safety. 

Recording the perspective of residents with dementia is often challenging as traditional qualitative research methods do not always fit PwD [16,17], and creative solutions are needed. In recent years, many such tools have been developed [18]. In their review of qualitative studies with PwD, Phillipson and Hammond [18] describe successful approaches that often utilise observation or involve doing something tangible, such as engaging participants by creating something or having them respond to a stimulus. This is reflected in the methodologies chosen in the current study. Firstly, we used observations. This method is one of the more commonly applied in dementia research [18,19] and has the obvious benefit that it is inclusive to all residents, no matter their cognitive and verbal capabilities. Second, we used focus group discussions. Even though this method does require a verbal response from participants (and thus was not suitable for the entire cohort), it has been successfully applied in previous studies with PwD [20,21]. Here, we chose to elicit a response from participants by presenting them with clear and concise statements about their lived experiences. Third, this study used the creative approach of creating mood boards to gain insights into participants who might not engage in conversation easily [22]. The rationale behind this was that the process of creating mood boards might serve as a conversation starter, and even if residents experience trouble expressing themselves verbally, then the resulting mood boards can be considered as the person’s input. Lastly, we used a simplified interview approach, where instead of responding to questions, participants were presented with a visual stimulus (pictograms/icons). Pictures have been shown to be a useful tool in starting a conversation with PwD [23]. The rationale was that there would be no formal distinction between the interviewer and participant and could jointly build a conversation around the theme presented in the image. 

The current study assumes a phenomenological approach and aims to further the debate around determining an optimal balance between freedom and safety for PwD living in a nursing home by gauging the values and opinions of people within this cohort themselves. Traditional qualitative methods like interviews or focus groups are often not the best tools in this cohort because they do not provide the flexibility to tailor these tools to the respondent’s capabilities [16,17]. As such, we have implemented a set of creative qualitative methods that could be adapted to fit the needs of a wide array of residents in the psychogeriatric department of a local nursing home. We use these tools to answer our research question: *How do people living with dementia value their own ‘freedom’ and ‘safety’?*


## 2. Materials and Methods

### 2.1. Participants

Data collection took place in two phases, the first being between October 2022 and February 2023 and the second being between November and December 2023. This study took place at a residential aged-care facility in a rural town in the Netherlands, in a department catering to people with memory problems (psychogeriatric department). Throughout the study, 25 residents lived in this department, and two residents with mild memory loss joined in from the neighbouring somatic department. Due to the memory problems in this group and their day-to-day differences in the willingness to participate, we did not aim to include all residents, but the research was conducted among participants who were available and responsive on the days of data collection. As a result, some participants were interviewed multiple times using different tools, while other potential participants were interviewed only once or not at all. The total group of participating residents was N = 22. All participants were in the age group of 75 to 96 years old, including 21 women and 1 man. Permission to collect data was obtained by phone with the resident’s registered first contact and checked verbally at the start of each data collection measure that involved social interactions from the residents themselves (i.e., this was not conducted with observations so as not to affect the measures). As some participants were not capable of giving informed verbal responses, any indication of unwillingness to participate (e.g., if a person stayed in their personal living space or used body language/demeanour to indicate they wanted to be left alone) was considered prior to data collection.

### 2.2. Procedure

Prior to data collection, observations were made in collaboration with nursing home staff to determine the stage of dementia in which the resident was located. Residents were categorised into three stages: residents who no longer initiate conversation (but might still be able to respond to questions), those who can speak but not always logically, and those who can speak logically. These observations determined the most suitable creative assessment tool for the individual and the moment. For the two groups that still initiate conversation independently, we used observations, a group discussion about relevant statements, mood boards, and icon/pictogram responses. Once, during the preparatory phase of this project, we attempted an interview with someone in this group, which proved too complex and was not repeated. We used observations for people who had lost the ability to initiate conversation. This set of methods originates from the ‘starter kit for identifying wishes of people with a damaged brain’, which was internally developed in collaboration with the supervising university of the second author’s thesis work. The second author completed all data collection as part of her thesis work. There was no prior relationship between this investigator and the residents. It was left up to the discretion of the investigator to determine which method would be implemented on the day based on the residents’ availability, mood and cooperativeness. All interactions with residents and all data collection and analyses took place in Dutch. Parts of these interactions were only translated for the purpose of reporting during the phase of writing the manuscript.

### 2.3. Interview

The structured interview was conducted with one resident following an interview guide developed elsewhere (‘*boom van vrijheid en veiligheid*’ or ‘tree of freedom and safety’; https://www.artemea.nl/thema-vrijheid/ (accessed on 1 November 2023). This tool was found too complex to use with this resident, so it was not further employed afterwards. However, the results obtained from this tool were still considered in the overall research: for this resident, the prompts ‘freedom’ and ‘safety’ were strongly associated with the end of the Second World War. Consequently, in the other methods, the questioning approach was adjusted to focus more on the resident’s daily sense of freedom and safety.

### 2.4. Observations

The observations, which averaged between 20 and 30 min, occurred in the department’s communal spaces, either the indoor courtyard or a communal living room. They were conducted using an observation schedule (Appendix A) that included questions based on the desires expressed by residents in both their verbal and nonverbal behaviour. The investigator positioned herself in the background close to the residents so she could hear them but would not be engaged in conversation. Audio recordings were made, but only if this was feasible considering background noise. Main outcomes were recorded on the observation schedule, and audio recordings were used to supplement the observation schedule after the observation period. If any resident engaged her in conversation, she would give a short answer to try to end the conversation. Immediately after the activity, the investigator summarised any conversation during the observation period on the observation schedule (supplemented by the audio recordings). The completed observation schedules were further analysed in terms of what individual needs or values are reflected in the observed behaviour. Observations were performed by one investigator, and both authors discussed the interpretation of the observations until a consensus was reached [24].

### 2.5. Group Discussion

In a group activity, statements and questions were presented to engage residents in a conversation about freedom and safety. The investigator acted as a moderator, keeping the conversation going where required and continuing to the next topic when all involved were satisfied. This was performed using printed statements and questions shown to the residents (a set of cards with the following statements was used: ‘Family is important’, ‘I decide’, ‘I feel at home’, ‘I am taken care of’, ‘I have boundaries’, and ‘I have privacy’. Statements were presented in Dutch). When needed, follow-up questions were asked to prompt discussion. The audio was recorded during the discussion. Immediately after the activity, the researcher summarised the conversation. Due to the conversational nature of this tool, large amounts of (irrelevant) chatter were recorded along with the group’s responses to the statements. Because of this, transcription was performed non-verbatim, in which only discourse relevant to the proposed statements was transcribed word-for-word. Both authors used the non-verbatim transcripts to induce common themes from the responses; any discrepancies between authors were discussed until a consensus was reached.

### 2.6. Mood Boards

Participants were involved individually or in groups of three to create a mood board on ‘freedom’ and/or ‘safety.’ This involved pasting magazine clippings that illustrated the participant’s feelings toward the theme. The investigator sourced magazines that covered everyday topics like gardening, the outdoors, and nature. The investigator assisted the resident throughout the process of clipping and pasting. Conversations were audio recorded, and immediately after the activity, the investigator summarised the conversations that occurred while creating the mood boards. As with the group discussions, building a conversation was considered more important than efficiently collecting information, resulting in large amounts of chatter. Rather than completing a verbatim transcript for all recordings, conversations were re-listened, and relevant statements were written down and coded from the audio directly. Both authors checked the results of this procedure until they were satisfied that all main messages were represented in the results. 

### 2.7. Icons

With this tool, the investigator brought some icons printed on paper to serve as a visual prompt and conversation starter. The investigator and the participant sat down in a one-on-one setting similar to a traditional interview, but rather than asking questions, the investigator showed an icon and asked what associations the participant had with the particular theme. The icons used were visual representations related to gardening (a pot with flowers), nature (a tree), dinner (a plate with a setting sun), positive aspects of life (a thumbs-up), and faith (a church). The investigator used follow-up questions to respond to what participants were saying and keep the conversation going, or when no obvious follow-up question presented itself, the investigator asked a question like ‘What else do you associate with this image?’ or moved on to the next image. The conversations were audio recorded and summarised based on these recordings per icon so they could be analysed for common themes. 

## 3. Results

A flowchart of which tools were used for which residents is displayed in Figure 1. In total, 22 residents participated in 37 creative qualitative assessment tools. 

### 3.1. Observations

Two observations occurred during an activity involving piano playing, one during a choir performance, one observation conducted in the indoor courtyard without any specific activity and the last in a communal living room. Results from the observations are summarised in Table 1. Among residents who can still speak, connection and participation in activities emerged as the key themes that were emphasised. For residents who can no longer initiate conversation, it was noted that they have a greater need for security, connection, and comfort. During group activities, conversations among residents mainly revolved around social contacts (primarily family), going outdoors, freedom and independence, and engaging in daily activities.

### 3.2. Group Discussions

This session was conducted with a group of eight residents during a knitting activity. Through discussions based on statements (Table 2), it emerged that all participating residents value their families. Most residents feel at home at the nursing home, mentioning that it is cosy, there are things to do, and family members visit regularly. Two residents expressed that they did not feel at home because they come from a different town. It was challenging to determine whether residents felt like they could make decisions about their own lives. Some residents indicated that they cannot always make choices independently, and a few residents expressed frustration about this. 

### 3.3. Mood Boards

Three sessions of creating mood boards were conducted one-on-one with a resident. For one resident, it became apparent that freedom meant being allowed to go outside alone and having the trust of the nursing home staff. Safety meant that their immediate environment felt familiar and secure and that staff looked out for them. This resident enjoys the freedom to control their own life, and it was described as anxiety-inducing when their dementia led them to forget tasks, which undermines this independence. Another resident created a mood board portraying rural life. They expressed that life on the farm gave them a sense of peace and freedom, as there is always something to do. Additionally, living with their partner and being surrounded by other people provided a sense of security. For the last resident who had a one-on-one session, it was apparent that a healthy lifestyle, physical activity, and being together with other people were highly valued. This resident did not specifically comment on the concepts of freedom and safety. 

Two sessions were completed, and three residents joined together. In the first, one resident occasionally walked away and returned later, while the other two residents stayed at the table during the activity. Ultimately, no mood board was created as they were satisfied with the conversation. These participants discussed a list of points defining freedom and safety, such as autonomy, hobbies, feeling useful, health, privacy and their financial situation. They primarily associate freedom with doing and saying what they want, caring for themselves, and feeling free and good. Safety, for these residents, means looking out for each other, helping one another and being aware that physical abilities may diminish with age. 

In the other session with three participants, two residents actively participated in the activity, creating a positive atmosphere as they engaged in conversations about beautiful nature photos from magazines. One resident mainly read aloud the text from the magazines. A mood board was created with numerous nature photos. The residents identified love, companionship, happiness, family, and nature as the most significant aspects of their lives.

### 3.4. Icons

From conversations using icons, it appears that all participating residents enjoy or used to enjoy going outside for walks and bike rides, have an affinity for nature and flowers, and prefer to eat together with other people, especially with family. Six of these residents perceived themselves as having a family life, believing their partners were still alive, although their partners had passed away. Notably, one resident mentioned not being allowed to go outside alone because she thinks other people (such as the nursing home staff) do not trust her due to the risk of getting lost. Additionally, three other residents indicated that they no longer go to the markets (which they used to do) or do not go outside in the evenings anymore. Finally, three residents emphasised the importance of faith as a source of comfort and stability. They also expressed satisfaction with the quality of care they receive in their nursing home. The responses to the pictograms are summarised in Table 3.

## 4. Discussion

The current study aimed to further the debate around determining an optimal balance between freedom and safety for PwD. A set of creative qualitative assessments was used to best match the individual cognitive capabilities of the people in this cohort [16,17]. Indeed, it was found that the one tool which was outlined like a structured interview was not a good match with the cohort. Combined, it was found that residents still voiced insights into their freedom and safety. Freedom was defined as being surrounded by people who give warmth and positive energy, being able to participate in activities, going outside, making one’s own decisions and performing tasks independently. How people experience freedom differs per individual. In general, the observation-based methods mainly used with residents who speak less or speak illogically emphasised the role of warmth and connection in experiencing freedom. The logical speech group voiced a wish for independence, making decisions and wishing to go outside independently. Safety is experienced by having people around you who look out for you and experiencing a homely living environment with a good balance between busy and quiet areas. Residents further defined retaining cognitive and physical capabilities and being in control of their lives as related to a sense of safety. It is interesting to note that mainly the residents that were in the logical speech phase perceived a lack of freedom or safety, for example in their disappointment in not being able to go to the markets anymore, frustration about a lack of control of one’s own life, their wish to go outside independently, or their independence in general. 

The participants in this study generally valued being treated as a person and showed frustration when decisions regarding freedom and safety were made for them (e.g., not being allowed to go outside anymore or not being included in discussing medical decisions). We interpret these results to show that the PwD in our cohort generally appreciated the elements of care organised as person-centred care [25,26,27]. In healthcare, the staff’s main concern often lies with patient safety, possibly neglecting other needs experienced by PwD [15,28]. This cohort is perhaps more susceptible to this than other patient groups, as they are generally less well-versed in objecting when needs might not be met, which is a well-known trigger for challenging behaviours [29]. Here, we have shown that nursing home residents with dementia still have varying needs in terms of freedom, which need to be considered in the decision-making process about care and living arrangements. Furthermore, they valued the environment that made them feel at home, rather than being in a medical department, both in terms of freedom (being able to socialise whenever they want) and safety (being in a homely environment where people look out for you). These findings support the movement beyond focusing on illness alone and back towards a holistic approach to care and illness [30]. 

Future work could expand on the current study by implementing the findings from this study in the decision-making process around personal freedoms and safety for nursing home residents. This is a particularly interesting topic in light of emerging GPS systems which allow monitoring and tracking of PwD from a distance [31]. This technology would allow for a more individual approach to personal freedom and might allow some PwD to roam their neighbourhood freely. However, this technology is also subject to ongoing ethical debates as increased personal freedom comes at the cost of decreased privacy without having the opportunity for informed consent [32]. The current study shows that, with the right tools, people with dementia can voice their values and opinions on personal freedom, which could form the basis of the discussion around the implementation of these sorts of technologies. 

From our results, we can suggest a list of conversation topics that could be used to establish one’s personal preferences in the balance between freedom and safety: participating in activities, going outside, having a social environment, autonomy, independence, a pleasant living environment, physical and cognitive functions, and a social environment that looks out for you and truly knows the resident. Healthcare professionals could use creative assessments to determine personal values in each topic and make decisions. For example, if one resident greatly values a safe personal environment and experiences freedom in connection to others, one could make different choices compared to residents who value going outside and do not mind some physical risks. Autonomy in making decisions about daily life has been shown to positively influence the quality of life [33], so involving the residents in these deliberations might already positively affect their lives. 

Some limitations need to be considered with the current results. First and foremost are some constraints to the academic rigour of the methods. In some cases, we have deviated from the gold standard in qualitative research [24]. Whilst all the methods used were pre-planned, we accepted that the mood of the day would often determine whether a specific resident was willing to participate and, therefore, which method was employed. As this decision was left up to the investigator on the day, this introduces a level of subjectivity. Further, in using most of the tools described here, starting a conversation was considered more important than getting a directed, concise response. We consider this fundamental in our approach to working with PwD. However, it resulted in long audio recordings with many irrelevant features. Due to time constraints and internal deadlines, these were not always verbatim transcribed, but relevant features were more directly extracted. This also potentially introduces subjectivity. Ultimately, these factors leave a mark on the reliability of the methods. We do not, however, consider it to have a detrimental impact on the validity of the results. In terms of validity, distilling the key messages from the recorded data was the main challenge, which was conducted here using a consensus model between authors. In doing so, we are confident that we have accurately captured the perspective of our cohort, giving merit to this study.

A second note needs to be placed with the current study’s generalisability [34]. This study was conducted in a nursing home in a small town in the Netherlands, and cultural differences exist even with other care facilities within the same organisation. For example, faith was found to be an important theme in some assessments, which is an important part of life in this rural area in the Netherlands, which might not be generalised to bigger cities. However, in general, a good understanding of this cohort’s values and the assessment methods was developed. Therefore, we can recommend implementing these methodological tools in future research or practice. 

## 5. Conclusions 

The results of this study indicate that there is not one single answer to the main question of how PwD value their own freedom and safety. Individual needs for freedom differ between people. Some might be satisfied with the freedom to socialise and find closeness with other people, while others value open doors and the freedom to go outside. In terms of safety, values seem more consistent. A feeling of safety is experienced in a homely living environment where people are around to look after one another. It is apparent that people in this cohort still have distinct values on freedom and safety, and it is recommended that people themselves are involved as much as possible in the policies implemented around their own lives. The creative qualitative assessment tools can be used as a conversation starter to assess how people with dementia value their own freedom and safety.

## Figures and Tables

**Figure 1 healthcare-12-01412-f001:**
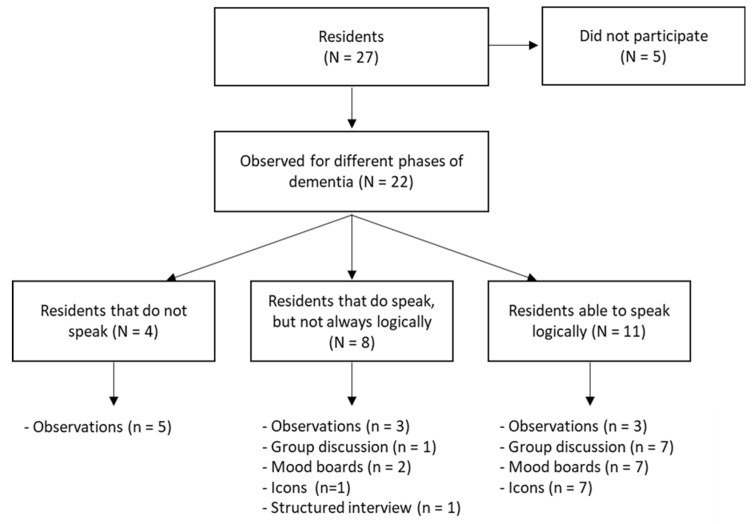
Overview of the number of participants (indicated with uppercase N) and participation in various measurement instruments (indicated with lowercase n). In some cases, more instruments were administered than in group N because participants were allowed to participate in multiple activities. In one instance, a resident changed dementia stages during the research period, from being able to speak but not logically to being unable to initiate conversation.

**Table 1 healthcare-12-01412-t001:** Results summary from the observations.

Situation	Dementia Phase	Observation	Interpretation
Resident attends a choir performance	Logical speech	Resident asks when the performance will end, how she will get home and whether someone could bring them	Behaviour shows a need for social contact, safety and security
Resident attends the weekly piano-play activity	Logical speech	Resident exclaims: “there are many empty seats”, pointing to the chairs next to them. Residents went to sit closer together. Resident asks: “have I seen you before?”	Behaviour shows a need for social contact, participation, quality of life and meaningfulness
Resident attends the weekly piano-play activity	Logical speech	Resident exclaims: “at home we also have a piano” and begins to talk about her husband playing	Behaviour shows a need for social contact, participation, appreciation and meaningfulness
Resident walking through the indoor courtyard	Illogical speech	Resident says: “I need to go home” and walks with a tense and searching demeanour	Behaviour shows a need for safety, security and social contact
Resident walking through indoor courtyard, at a certain point the fire alarm goes off	Illogical speech	Fire alarm led to scared response: “what was that?”. Resident walks with searching demeanor. Later, near dinnertime, the resident asks passersby where her children are	Behaviour shows a need for safety, security, social contact and meaningfulness
Resident attends the weekly piano-play activity	No speech	Resident moves their feet with the music. Resident seemed activated and ate better than usual	Behaviour shows a need for quality of life, meaningfulness and safety
Resident attends the weekly piano-play activity, a volunteer is stroking their arm	No speech	Resident’s demeanour and facial expression got less tense	Behaviour shows a need for quality of life, meaningfulness, social contact and safety
Resident is sitting at a table in the living room	No speech	Resident’s demeanor was quiet. Occasionally, the resident gibbered some words that sounded satisfied	Behaviour shows a need for safety
Resident attends a choir performance	No speech	Resident seemed stimulated to occasionally move the arms and legs	Behaviour shows a need for quality of life, meaningfulness and safety

**Table 2 healthcare-12-01412-t002:** Responses to specific statements in group discussion.

Statement	Response Summary
Family is important	- Yes! I agree - It is good to have a family - They do not abandon you
I decide	- It depends on the topic - This is a tough question - Not always, if I do not care - No, I am annoyed that I am not allowed to decide for myself
I am/feel at home	- I enjoy the atmosphere; there is always something to do - You do not have to sit down doing nothing - I do not feel at home here because I do not live here. You know that - No, I am not from here - Yes, my family often visits - Yes, it is good here
I am taken care of	- What does that mean? (Some initial debate emerged about the meaning of this statement) - Yes, I think so. They help us
I have boundaries	- Yes - Angry people are unpleasant - ‘To here and no further’—I do recognise this
I have privacy	- Sometimes, I do not come here often - Yes, if I want to be alone, I can go to my own room

**Table 3 healthcare-12-01412-t003:** Outcomes from discussing the icons.

Icons	Dementia Phase	Response Summary
Tree	Logical speech group	- I enjoy going outside - I love nature - I enjoy walking and riding a bike - Movement is important - I no longer go to the markets
	Illogical speech group	- <looks at daughter for confirmation> I am not allowed to go outside
Dinner	Logical speech group	- Sharing a meal with family - We eat a hot meal at mid-day
	Illogical speech group	- Sharing a meal with family
Gardening	Logical speech group	- I enjoy gardening outside - I enjoy having flowers out on the table
	Illogical speech group	- I love flowers - Flowers out in the garden are pretty
Thumbs up	Logical speech group	- I receive a good standard of care - I would have preferred to live at home with my partner
	Illogical speech group	- I am fine
Church	Logical speech group	- Church is very important. It gives comfort - I was raised with the church. I got that from my parents
	Illogical speech group	- I go to church - Church is very important

## Data Availability

Data are contained within the article.

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
