# Peer review of "From the Perspective of People with Dementia: Using Creative Qualitative Measures to Assess the Values and Opinions on Freedom and Safety among People Living with Dementia"

_healthcare, 2024, doi:10.3390/healthcare12141412_

Round 1

Reviewer 1 Report

Comments and Suggestions for Authors

From the perspective of people with dementia: using creative qualitative measures to assess the values and opinions on freedom and safety among people living with dementia

Introduction

The study aim was to support the debate around determining an optimal balance between freedom and safety for seniors with dementia living in a nursing home.

Material and methods

The authors used a set of qualitative methods: observation, interview. group discussion, mood boards and icons.

Results

During group activities, conversations among residents mainly revolved around social contacts, going outdoors, freedom and independence and engaging in daily activities.

Through discussions based on statements it emerged that all participating residents value their families.

The residents identified love, companionship, happiness, family, and nature as the most significant aspects of their lives.

Discussion and conclusions

Using all tools, it was found that residents still voiced insights into their freedom and safety.

The mainly the residents that were in the logical speech phase perceived a lack of freedom or safety, their independence in general.

From results, authors can suggest a list of conversation topics that could be used to establish one’s personal preferences: participating in activities, going outside, having a social environment, autonomy, independence, a pleasant living environment, physical and cognitive functions, a social environment that looks out for you and truly knows the resident.

Recommendations:

I recommend adding other studies with the examined issue to the discussion.

Older literary sources were used in the article, for example:  2006, 2011, 2013 and others.

Accept after minor revision (corrections to minor)

Author Response

Recommendations:

I recommend adding other studies with the examined issue to the discussion.

Older literary sources were used in the article, for example:  2006, 2011, 2013 and others.

Response:

We thank the reviewer for their assessment of the manuscript and for their recommendation. Expanded on the discussion and, in doing so, included more recent studies in this section. 

Reviewer 2 Report

Comments and Suggestions for Authors

Thank you for your paper. I enjoyed reading it, and am delighted that you are introducing diverse ways of gathering data.

The abstract was clear, as was the background. The experience of undertaking the interview was an interesting one, and it provides great insight into research with this population of PwD. As you correctly point out in your introduction, there is an expected global rise in the numbers of people living with dementia, and so being able to accurately interpret the wishes and beliefs of this group of people, is, and will be, vital to providing person-centred care.

This is potentially an important paper, and so I would welcome a lot more detail in your methods and analysis sections (see some suggestions below). I suggest you align these sections to a quality framework such as COREQ (Consolidated criteria for reporting qualitative research), as this will strengthen both the quality of the paper, and its reach and value to those interested in progressing methodology. The reader needs to clearly understand the rigour involved in each of these different techniques, and also have enough information to learn, and possibly try these techniques themselves.

The discussion included some useful insights. This section would be strengthened by linking your findings to the wider literature, especially in terms of research with participants who are living with cognitive challenges.

The reference list was relevant, and I would have liked to see more citations related to your methods and analysis.

Some comments:

The different types of methods are very interesting. I think the section would benefit from much greater detail around how you went from (for example) ‘observations’ to your interpretations/findings. I have outlined some ideas for consideration (below).

Did the researcher have a topic guide (and I mean this is a broad sense, as you already articulated how the more formal interview was not possible)? Who gathered the data (at the end you state that the second author did so – thank you, however, perhaps include some further relevant background), and did they undertake any reflexivity exercise? What was their relationship with the study participants?

What language was used? How were the data managed in terms of transcription (and translation if a different language other than English was used).

Please give a sense of how involved the researcher was within each of the data gathering activities. For example, with the mood board – where did you source the magazine clippings? What was the role of the researcher? For example, did the research help identify potential artefacts, or help cutting out images that the participant identified…?

It is clear that for much of the gathering data events, that others were present. Were their opinions, beliefs, verbal/non-verbal data included in the interpretation of data?

Did you audio and video-record everything, or some of the interactions? How did you analyse verbal data? How did you analyse non-verbal data, e.g. movements (from facial expressions to more global movement like walking and hand gestures)? For example, did you have a non-verbal framework as a guide, such as proxemics, or kinesics? Another example (Line 178): “it was apparent…” How did you know? If it was apparent to one researcher, how did you check that the rest of the team agreed? Was there discussion and debate related to interpreting all your data, and if so, how was conflict resolved?

Some minor editing issues

Line 105: what conversations? The previous sentences referred to people who couldn’t speak, so for these participants, you mentioned observations. Further on, you clarify what you mean by observations (Lines 115+), so I would either re-phrase the sentence or (re)move it.

Table 1: row 4, column 3. Do you mean “ we also have a piano”…?

Table 2: row 2, line 2: Do you mean that this is a tough question?

Table 2: row 4: I don’t see the link between the statement “I am being taken care of.“ and the response summary “What does that mean?”

Line 247: separate words: choices compared to

Author Response

Comment: This is potentially an important paper, and so I would welcome a lot more detail in your methods and analysis sections (see some suggestions below). I suggest you align these sections to a quality framework such as COREQ (Consolidated criteria for reporting qualitative research), as this will strengthen both the quality of the paper, and its reach and value to those interested in progressing methodology. The reader needs to clearly understand the rigour involved in each of these different techniques, and also have enough information to learn, and possibly try these techniques themselves.

Response:

We thank the reviewer for assessing the manuscript and are happy to read that some key components of the narrative have been well received. We thank the reviewer, in particular, for this comment. Both authors have limited experience with qualitative research, so a constructive comment like this one is greatly valuable to us. We have now used the SRQR guideline [1–3] to improve the reporting standard of the manuscript. This has resulted in numerous changes throughout the manuscript, with major changes:

  • Specification of the research approach (line 94 )
  • More details on data collection period and sampling (lines 105-107 & 121-124)
  • Added researcher characteristics/relationships (lines 137-138)
  • Added details of used instruments (throughout the methods section and sup. file 1)
  • A reflection on the limitations of our methods in the discussion (line 342-358)

Cited works:

  1. O’Brien, B.C, et al. Academic Medicine 2014, 89, 1245–1251, doi:10.1097/ACM.0000000000000388.
  2. Dossett, L.A.; et al. A.. JAMA Surg 2021, 156, 875–876. Doi: 10.1001/jamasurg.2021.0525
  3. Buus, N.; Perron, A. Int J Nurs Stud 2020, 102, doi:10.1016/j.ijnurstu.2019.103452.

Comment: The discussion included some useful insights. This section would be strengthened by linking your findings to the wider literature, especially in terms of research with participants who are living with cognitive challenges.

Response: Thank you for this comment. We have added two parts to the discussion linking it to relevant research topics. Firstly by expanding on the needs of people with dementia (line 298-305) and secondly by discussing what these results mean in light of another big change in dementia care policy, the use of tracking technologies such as GPS (line 313-321)

Combined comments on methods section:

  • The reference list was relevant, and I would have liked to see more citations related to your methods and analysis.
  • The different types of methods are very interesting. I think the section would benefit from much greater detail around how you went from (for example) ‘observations’ to your interpretations/findings. I have outlined some ideas for consideration (below).
  • Did the researcher have a topic guide (and I mean this is a broad sense, as you already articulated how the more formal interview was not possible)? Who gathered the data (at the end you state that the second author did so – thank you, however, perhaps include some further relevant background), and did they undertake any reflexivity exercise? What was their relationship with the study participants?
  • What language was used? How were the data managed in terms of transcription (and translation if a different language other than English was used).
  • Please give a sense of how involved the researcher was within each of the data gathering activities. For example, with the mood board – where did you source the magazine clippings? What was the role of the researcher? For example, did the research help identify potential artefacts, or help cutting out images that the participant identified…?
  • It is clear that for much of the gathering data events, that others were present. Were their opinions, beliefs, verbal/non-verbal data included in the interpretation of data?
  • Did you audio and video-record everything, or some of the interactions? How did you analyse verbal data? How did you analyse non-verbal data, e.g. movements (from facial expressions to more global movement like walking and hand gestures)? For example, did you have a non-verbal framework as a guide, such as proxemics, or kinesics? Another example (Line 178): “it was apparent…” How did you know? If it was apparent to one researcher, how did you check that the rest of the team agreed? Was there discussion and debate related to interpreting all your data, and if so, how was conflict resolved?

Response:

With these comments collectively, the reviewer indicates a clear need for more clarity in the backgrounds of the methods section and the methods section itself. Taken together with the earlier comment on using reporting guidelines, and comments made by other reviewers, we have chosen to make large changes to boost transparency. As such, we have: i. added the background of the methodologies to the introduction (line 66-89), ii. added more details on the circumstances and setting of data collection (throughout methods) and iii. further specified the protocols used (throughout methods section).

On the specific questions postulated by the reviewer, we respond:

  • Citations have been added, mainly to the background section (line 66-89)
  • The link between ‘raw’ results and conclusions has been made more apparent thanks to a better description of the protocols, so the reader is informed by which steps guided the thinking process
  • Further details have been added about the data collection, such as the timescale (line 104-106), role and relationship of the researcher (line 119-123, line 137-140), and how consensus was reached between authors (throughout methods, per section).
  • Details on language have been added (line 140-142)
  • Involvement of the investigator is now described in more detail per methodology.
  • If others were of influence on the data collection, details were included in the results (for example, in the observation results, interaction with a volunteer is described: Table 1 row 7)
  • Details on the protocol were added about data analysis, including the use of audio recordings and transcription. We did not use a specific guide for nonverbal communication but rather relied on the researcher's experience collecting the data, which was later discussed and corroborated with the other researcher, so both researchers agreed with the conclusions.

Comment: Line 105: what conversations? The previous sentences referred to people who couldn’t speak, so for these participants, you mentioned observations. Further on, you clarify what you mean by observations (Lines 115+), so I would either re-phrase the sentence or (re)move it.

Response: We agree with the confusion in this statement and have removed it.

Comment: Table 1: row 4, column 3. Do you mean “ we also have a piano”…?

Response: Indeed, this is exactly what was meant. Thank you for picking up on this!

Comment: Table 2: row 2, line 2: Do you mean that this is a tough question?

Response: Thank you for thoroughly reading the manuscript. We really appreciate this in-depth feedback.

Comment: Table 2: row 4: I don’t see the link between the statement “I am being taken care of.“ and the response summary “What does that mean?”

Response: There was some initial debate about the meaning of the question, which we summarised with this response. We have added a short explanation to this extend to table 2.

Comment: Line 247: separate words: choices compared to

Response: Thank you again for taking the time to provide feedback in this level of detail.

Reviewer 3 Report

Comments and Suggestions for Authors

Dear Editor,

Thank you for the opportunity to review this manuscript. This manuscript reports the findings of a qualitative study using creative measures to investigate values and opinions on freedom and safety among people living with dementia. The manuscript is well read, and the findings can be useful to expand various methods for obtaining data in the dementia research area. However, this manuscript needs improvements, especially in the method part, to enhance its clarity. Below are our comments and suggestions:

  1. Were there any previous studies in the literature that have used these tools? Please explain and add to the background part.
  2. On page 3, lines 120–121, it is written that “outcomes were derived from recordings of the conversation and the observation schedules. How was the recording of the observation conducted? Was the observation audio visually recorded? Or is only audio recorded? Please explain.
  3. Please add more explanations about the activities in the icons. It is not clear what activities were provided when collecting the data with icons. Did authors create particular activities, such as taking the participants to the church or showing a picture of the church and having a conversation about church? How about other icons, such as a thumbs-up, a tree, a plate with a setting sun, and a pot with flowers? What did the researcher actually do in this part? Please explain.
  4. Please add an explanation about the inclusion criteria for the participants. On page 2, line 84, it is written that participants were those who were available and responsive during the days of data collection. However, it is also mentioned further that participants include those who did not speak. Could you please clarify the criteria for responsiveness? How do we know that those who did not speak are also responsive?
  5. When was the data collection period? Which month and year? Please add.
  6. Please give the interview or group discussion guide that lists the questions being asked.
  7. In the material and method sections, please provide a sub-section for data analysis that explains how authors analyzed the data using various methods, including observations, interviews, group discussions, mood boards, and icons.
  8. How did researchers maintain the rigor and trustworthiness of this study? How did the authors ensure that the interpretation of observations, mood boards, icons, and summary of the group discussion were relevant to the data? Please explain.
  9. Please check the whole manuscript and fix some typing errors, for example, on page 2, line 65, “balance between freedom an safety.” Do you mean freedom and safety?

Author Response

Comment: Were there any previous studies in the literature that have used these tools? Please explain and add to the background part.

Response: We thank the reviewer for their detailed feedback on our manuscript. On this point, we have added further details and reference to previous studies to the background section (line 66-89)

Comment: On page 3, lines 120–121, it is written that “outcomes were derived from recordings of the conversation and the observation schedules. How was the recording of the observation conducted? Was the observation audio visually recorded? Or is only audio recorded? Please explain.

Response: Thank you for the opportunity to improve the transparency in our methods section. We have added more details to the protocol description for each of the implemented methods. We recorded the observations using field notes on the observation schedule (added as a sup. file) and additionally audio recordings when the background noise allowed for this (this was very limited however, as some observations took place during piano play and choir performance).

Comment: Please add more explanations about the activities in the icons. It is not clear what activities were provided when collecting the data with icons. Did authors create particular activities, such as taking the participants to the church or showing a picture of the church and having a conversation about church? How about other icons, such as a thumbs-up, a tree, a plate with a setting sun, and a pot with flowers? What did the researcher actually do in this part? Please explain.

Response: More details have been added to the icons activity. We are under the impression that it might not have been clear that these were merely visual representations of the themes to start the conversation. As such, we have emphasised this stronger in the methods section (line  188-198)

Comment: Please add an explanation about the inclusion criteria for the participants. On page 2, line 84, it is written that participants were those who were available and responsive during the days of data collection. However, it is also mentioned further that participants include those who did not speak. Could you please clarify the criteria for responsiveness? How do we know that those who did not speak are also responsive?

Response: With ‘responsive’ we meant any verbal or non-verbal response to the interaction. However, we see that this is not clear enough in the previous version of the manuscript and have thus added more details to our approach (line 119-123).

Comment: When was the data collection period? Which month and year? Please add.

Response: Details have been added to the manuscript (line 104-106).

Comment: Please give the interview or group discussion guide that lists the questions being asked.

Response: For the interview, we used guidelines available through the link in the manuscript. As this was a pre-developed tool, we unfortunately cannot share it here due to copyright constraints. For the group discussion, all statements that served as inputs have now been added to the appropriate paragraph in the methods section. Finally, we have also added the observation schedule as a supplementary file.

Comment: In the material and method sections, please provide a sub-section for data analysis that explains how authors analysed the data using various methods, including observations, interviews, group discussions, mood boards, and icons.

Response: We thank the reviewer for this idea, as it will greatly help improve the clarity of the protocols. We did, however, opt not to designate a separate paragraph to this end, but rather to integrate this into the existing sections per methodology. As each methodology comes with its own data analysis protocol, we considered this to be a clearer way of presenting this information.

Comment: How did researchers maintain the rigor and trustworthiness of this study? How did the authors ensure that the interpretation of observations, mood boards, icons, and summary of the group discussion were relevant to the data? Please explain.

Response: In answering this and earlier queries about the reporting of the methods, we have reflected on the rigour of our methods in general. Compared to the standards in the field, we realise that some decisions in our protocols might be considered limitations to the study. We consider it important to transparently present these limitations in the discussion section (lines 342-358). This way, the reader can fairly gauge the quality of the results.

Comment: To increase trustworthiness, we have installed a consensus model in which both authors checked and discussed all crucial steps in interpreting the results. We have now added statements describing this approach to the relevant parts in the methods section.

Comment: Please check the whole manuscript and fix some typing errors, for example, on page 2, line 65, “balance between freedom an safety.” Do you mean freedom and safety?

Response: Thank you for this input. We have carefully proofread the manuscript again and fixed the typing errors.

Round 2

Reviewer 2 Report

Comments and Suggestions for Authors

Thank you for your detailed responses to my comments, and for your hard work in amending this paper. There is one small editing error (line 180), and I think a verb is missing. Aside from that, this is now a lot clearer.

I wish you every success in your ongoing research.

Author Response

Thank you for your continued review of our manuscript. We have amended this sentence to make it work.